# Ecogeography of *Dioscorea remotiflora* Kunth: An Endemic Species from Mexico

**DOI:** 10.3390/plants12203654

**Published:** 2023-10-23

**Authors:** Jocelyn Maira Velázquez-Hernández, José Ariel Ruíz-Corral, Noé Durán-Puga, Miguel Ángel Macías, Diego Raymundo González-Eguiarte, Fernando Santacruz-Ruvalcaba, Giovanni Emmanuel García-Romero, Agustín Gallegos-Rodríguez

**Affiliations:** 1Department of Agricultural Production, CUCBA, University of Guadalajara, Cam. Ramón Padilla Sánchez 2100, Las Agujas, Zapopan 45110, Jalisco, Mexico; jocelynv795@gmail.com (J.M.V.-H.); noe.duran@academicos.udg.mx (N.D.-P.); diego.geguiarte@academicos.udg.mx (D.R.G.-E.); fernando.santacruz@academicos.udg.mx (F.S.-R.); 2Department of Environmental Sciences, CUCBA, University of Guadalajara, Cam. Ramón Padilla Sánchez 2100, Las Agujas, Zapopan 45110, Jalisco, Mexico; mmacias@cucba.udg.mx; 3Environment Department of the Municipality of Guadalajara, Av. Miguel Hidalgo y Costilla 426, Downtown, Guadalajara 44100, Jalisco, Mexico; geog.vanni@gmail.com; 4Departmento de Producción Forestal, CUCBA, University of Guadalajara, Cam. Ramón Padilla Sánchez 2100, Las Agujas, Zapopan 45110, Jalisco, Mexico; agustin.gallegos@academicos.udg.mx

**Keywords:** *Dioscorea remotiflora*, Mexican endemic species, niche modeling, ecological descriptors, climatic adaptation

## Abstract

*Dioscorea remotiflora*, a perennial climbing herbaceous plant native to Mexico, produces tubers with great nutritional and ethnobotanical value. However, most ecological aspects of this plant remain unknown, which limits its cultivation and use. This is why the objective of this research was to characterize the ecogeography of *D. remotiflora* as a source to determine its edaphoclimatic adaptability and current and potential distribution. A comprehensive database encompassing 480 geo-referenced accessions was assembled from different data sources. Using the Agroclimatic Information System for México and Central America (SIAMEXCA), 42 environmental variables were formulated. The MaxEnt model within the Kuenm R package was employed to predict the species distribution. The findings reveal a greater presence of *D. remotiflora* in harsh environments, characterized by arid to semiarid conditions, poor soils, and hot climates with long dry periods. Niche modeling revealed that seven key variables determine the geographical distribution of *D. remotiflora*: precipitation of the warmest quarter, precipitation of the driest month, minimum temperature of the coldest month, November–April solar radiation, annual mean relative humidity, annual moisture availability index, and May–October mean temperature. The current potential distribution of *D. remotiflora* is 428,747.68 km^2^. Favorable regions for *D. remotiflora* coincide with its current presence sites, while other suitable areas, such as the Yucatán Peninsula, northeast region, and Gulf of Mexico, offer potential expansion opportunities for the species distribution. The comprehensive characterization of *Dioscorea remotiflora*, encompassing aspects such as its soil habitats and climate adaptation, becomes essential not only for understanding its ecology but also for maximizing its economic potential. This will enable not only its sustainable use but also the exploration of commercial applications in sectors such as the pharmaceutical and food industries, thus providing a broader approach for its conservation and optimal utilization in the near future.

## 1. Introduction

*Dioscorea remotiflora* is a wild, monocotyledonous, perennial, and climbing plant, with cordate leaves, dioecious flowers, and seeds in axillary clusters. It belongs to the *Dioscoreaceae* family [1] and is one of the native species of Mexico from the genus *Dioscorea* [2,3]; currently, it is considered an endemic species of Mexico [4]. Although *D. remotiflora* is not a cultivated plant, its tubers have been collected since prehistoric times to be used as food [5]; they contain 85% carbohydrates, 7.35% proteins, 3.76% lipids, and 3.68% minerals (K, Fe, Na, and Mg). Therefore, it is used as a healthy snack or even as a gourmet dish [6,7]. The tubers contain valuable secondary metabolites such as steroids, saponins, and diosgenin, but it is assumed that they may contain other compounds for medicinal use; however, this and other aspects of this plant remain unknown; thus, *D. remotiflora* is currently considered understudied and underutilized [7].

*D. remotiflora* is mainly distributed in the central, southern, and western regions of Mexico [8], generally in dry deciduous tropical forests [9,10], which indicates its adaptive capacity to diverse environments [11], including unfavorable edaphoclimatic conditions for agriculture, such as poor soils and semiarid lands [9,10]. Species of the Dioscorea genus, related to *D. remotiflora*, have even shown resistance to water and saline stress [12]. Nevertheless, the adaptive capacity of this plant to different abiotic environments has not been studied in depth either.

According to previous reports on *Dioscorea* species [13] and preliminary data and their analysis, *D. remotiflora* could constitute an excellent alternative crop and food for regions where climate change is imposing increasingly adverse environmental conditions for agriculture. Nowadays, plants with these characteristics are being demanded all over the world to better face climate change’s effects on agricultural lands [14].

Up to now, the reports regarding the description of the ecology, climatic adaptation, and potential distribution of *D. remotiflora* are bare and insufficient; this hinders the design of strategies for its conservation and optimal use [15,16]. In order to achieve such purposes, species distribution modeling (SDM) and characterization of the species’ ecogeography are usually appropriate tools [17]. Therefore, the objective of this research was to use the occurrence data of *D. remotiflora*, in its natural distribution areas, to conduct an ecogeographical analysis in order to elucidate the contribution of several ecological descriptors in determining its current distribution, identify its adaptation patterns, and develop optimal MaxEnt models of potential geographical distribution, through the use of the Kuenm R package, which automates the creation, calibration, and evaluation of ecological niche models [18].

Plant growth and distribution are the results of the species’ response to the environmental complex that prevails in the occurrence sites [19,20], where aspects of climate, soil, vegetation, and others concur. As a species moves from its center of origin to other geographic regions, it finds it necessary to adapt to different environmental conditions. If the adaptation process is successful, then the species will have colonized new territories, extending its distribution and, thus, its adaptation environmental scope, which usually triggers an increase in its tolerance to abiotic and biotic stress [21]; this aspect may not be manifested uniformly in all the ecogeographic populations of the species [22]; this is why it is relevant to consider all the accessions and populations of the species under study.

## 2. Results

### 2.1. Selection of Environmental Variables

A Spearman correlation analysis enabled us to reduce the environmental variables to be used in further analysis from 42 to 20 [23,24]. Furthermore, preliminary tests with MaxEnt revealed that out of the twenty variables, seven are the most relevant in determining the distribution of *D. remotiflora*.

### 2.2. Current Distribution, Climatic Adaptation, and Ecological Descriptors

Figure 1 shows the current distribution of *D. remotiflora* in the agroclimatic regions of Mexico; as shown, this species is found predominantly in the areas near the Mexican Pacific coast, mainly in the central and southern portions. A greater presence of *D. remotiflora* is observed in the following agroclimatic regions: humid–subhumid semiwarm (98 accessions), dry–subhumid semiwarm (88 accessions), humid–subhumid warm (67 accessions), and dry–subhumid warm (66 accessions). However, *D. remotiflora* is present in 15 of the 29 agroclimatic regions of Mexico (Table 1), which enables this species to develop in thermal zones from semicold to very warm and in hydric zones from semiarid to humid.

Table 2 shows the FAO soil units and the textural classes in which *D. remotiflora* is distributed. As can be seen, most of the occurrence sites of this species are distributed in the following soil types: Lithosol (144 accessions), Calcaric Regosol (98 accessions), Eutric Regosol (99 accessions), and Haplic Faozem (45 accessions) (Figure 2).

The selected ecological descriptors provide essential information for understanding the environmental requirements of *D. remotiflora* and its adaptation to a range of climate and soil conditions. This is crucial for its conservation and for identifying optimal areas for its cultivation or preservation in the wild. Table 3 shows the ecological descriptors of *D. remotiflora*, which contain the environmental adaptation ranges (RAA) of the species (minimum and maximum values of environmental variables that allow the presence of *D. remotiflora*) and the optimal environmental range (RAO), which allows the higher frequency of occurrence sites of this plant. According to the information in Table 3, *D. remotiflora* is distributed in environments with an annual moisture availability index (MAI) ranging from 0.27 to 2.32, with an optimal range of 0.40 to 0.99, which corresponds to an RAA and RAO of an annual precipitation interval of 444 to 2886 and 700 to 1299 mm, respectively. This plant prefers areas where precipitation in the wettest quarter ranges from 400 to 884 mm, although it tolerates conditions from 240 to 1204 mm. This species can tolerate a long season (November–April) with low precipitation, even with an accumulation of only 23 mm in those six months (Table 3). The optimum growing season goes from 120 to 150 days, although it grows in regions with a growing season as long as 120–190 days.

With respect to height, *D. remotiflora* is present from 6 to 4295 masl, but most of the presence sites occur between 200 and 800 masl. This encourages this species to develop in areas with an average annual temperature between 14.7 and 28.5 °C, with an optimum of 19 to 27 °C, an extreme monthly average maximum temperature of 41.2 °C (maximum maximorum temperature), and an extreme monthly average minimum temperature of 1.7 °C (minimum minimorum temperature). In both the seasonal periods May–October and November–April, the interval in which the greatest number of accessions occurs is from 19 to 26 °C, very similar to the optimum average annual temperature. Regarding soil properties, *D. remotiflora* is present in coarse-, medium-, and fine-textured soils, but its presence is more abundant in coarse-textured soils; the number of occurrence sites decreases markedly in medium- and fine-textured soils, indicating a clear preference of the species for soils with excellent drainage.

### 2.3. Modeling Distribution Niches of D. remotiflora

Due to MaxEnt not being exempt from the effects of collinearity and the fact that these can intervene in the estimation of the factors, as well as inferring the uncertainties when the models are transferred spatially and temporally [25], and with the purpose of obtaining an accurate model with a reduced number of variables, eliminating the possibility of obtaining an over-fit or over-parameterized model [18], the Kuenm R package allowed, through ecological niche modeling, the optimization of the number of environmental variables, leaving only seven, due to their greater contribution to the presence and distribution of *D. remotiflora*. The results of the Jackknife statistical test of factorial importance reported that the most determining environmental variables in the presence and distribution of *D. remotiflora* are the precipitation of the warmest quarter (42.4%), mean precipitation of the driest month (17.5%), minimum temperature of the coldest month (15%), November–April mean solar radiation (10%), annual mean relative humidity (8.5%), annual moisture availability index (5.7%), and May–October mean temperature (0.9%) (Table 4).

Based on the results obtained through the Jackknife test, three sets of cases were analyzed: “only with variable”, “without variable”, and “with all variables” [26], thus revealing the effects of environmental variables in the appropriate range of *D. remotiflora* (Figure 3). Among these environmental variables, it was found that solar radiation from November to April was the most relevant, with a regularized training gain greater than 0.8. In addition, other variables of significant importance were identified, such as the precipitation of the warmest quarter, the annual humidity index, relative humidity, the minimum temperature of the coldest month, the precipitation of the driest month, and the average temperature from May to October. In all these cases, the regularized training gains exceeded 0.6.

The Kuenm R package enabled us to obtain 372 models (Figure 4), all of them significant; however, only two models positively met the Akaike criterion (AIC = 0) and a maximum omission rate of 5%. The Kuenm R package implemented with MaxEnt determined that 3.0 was the optimum regularization multiplier. The model finally selected to depict the distribution of *D. remotiflora* was judged as excellent since the AUC of the ROC curve accounted for 0.935 (Table 5).

Figure 5 shows the current and potential distribution of *D. remotiflora* based on thresholding the environmental suitability with the “Balance training omission, predicted area, and threshold value” criterion. The map in Figure 4 shows four types of areas: areas not suitable (white color), areas with low environmental suitability (light green), areas with medium suitability (dark green), and areas highly suitable for *D. remotiflora* presence (dark grey), which matches the largest number of occurrence sites. The total potential distribution area for *D. remotiflora* accounted for 428,747.68 km^2^, distributed as follows: The areas with low suitability account for 150,547.104 km^2^ and are located in Sonora, north Sinaloa, Guanajuato, Querétaro, Mexico State, coasts of Guerrero, central Oaxaca, Chiapas, Campeche, San Luis Potosí, south Tamaulipas, and Veracruz. The areas with medium suitability cover 127,817,096 km^2^ and are located in southern Sonora, southern Chihuahua, northern Sinaloa, the coasts of Nayarit, Jalisco, Michoacán, Guerrero, Oaxaca, and Chiapas. The highly suitable areas cover 150,383.48 km^2^ (dark gray) and are located in Sinaloa, Nayarit, Jalisco, Colima, Michoacán, Morelos, little portions of Puebla and Tamaulipas, northern Guerrero, and the coasts of Oaxaca. As can be seen in Figure 4, there are potential areas in regions where the presence of *D. remotiflora* has not yet been reported, which indicates territories that could be the target of future exploration of new populations. This is the case for the Peninsula of Yucatán, central Veracruz, southeastern San Luis Potosí, southern Tamaulipas, Guanajuato, northeastern Jalisco, northern Sinaloa, northern Hidalgo, and western Mexico State.

## 3. Discussion

### 3.1. Current Distribution, Climate Adaptation, and Ecological Descriptors

Most of the *D. remotiflora* occurrence sites are concentrated amongst the dry–subhumid and humid–subhumid agroclimatic regions with the warm and semiwarm variants, with an annual mean temperature between 19 and 27 °C and an annual moisture availability index (MAI) between 0.4 and 0.99 (Table 4). These results agree with the ones reported by [2] who indicate that *D. remotiflora* is distributed in areas with semiwarm, warm, and subhumid climates, ranging from the northern portion to the central region of Mexico. However, an interesting aspect is the climatic extremes in which *D. remotiflora* can adapt, meaning that even when most of the occurrence sites of this species are located between 700 and 1299 mm of annual rainfall, there are populations that subsist with 444 mm per year, and on the other extreme, other populations subsist in sites with 2886 mm of precipitation per year. When combined with annual potential evapotranspiration data, these precipitation values translate into MAI values from 0.27 to 2.32, which according to the arid zones scheme from the UNEP [27] (adapted by Ruiz et al., 2004) match with semiarid to very wet lands. Regarding this, [28] mentions that some species of *Dioscorea* adapt to dry periods and can survive under conditions of water deficit better than many other species and crops. This seems to be the case for *D. remotiflora*, according to the ecological indicators obtained. The presence of populations of *D. remotiflora* in extremely humid sites (2886 annual mm and MAI > 2) could be explained in sites with excellent soil drainage, or these data can be considered as an indicator of the ability of this species to colonize habitats typically unsuitable for its development [9].

Regarding the variable precipitation of the warmest quarter, which was the most significant for the presence of *D. remotiflora* in the MaxEnt modelling, Table 4 indicates a range of 240 to 1204 mm, with an optimal range of 400–884 mm. Considering then that *D. remotiflora* is present in sites with a minimum annual rainfall of 444 mm and that, from those millimeters, at least 240 must occur in the warmest quarter [29], we can conclude that this plant has the ability to adapt and develop in environments with an irregular distribution of precipitation, which may also represent a comparative advantage of *D. remotiflora* in relation to other species. This also leads us to conclude that *D. remotiflora* is a species that adapts more to moderate-to-low humidity conditions; thus, very humid or arid environments are not conducive to the high productivity of this species. The results obtained also coincide with previous reports related to the adaptation of the genus *Dioscorea* to tropical and subtropical zones, with tolerance of some species to conditions of water deficit [13,28].

The environmental characterization of the presence sites of *D. remotiflora* shows that there is a greater number of accessions of this plant in Lithosol, Calcaric Regosol, and Eutric Regosol soils, which do not offer the best conditions for the development of vegetation and crops [30]. According to [31], Leptosols (Litosols and Rendzinas) represent 28.3% of the Mexican territory and are characterized by very thin, stony, and poorly developed soils that can contain a large amount of calcareous material, which immobilizes mineral nutrients. They are common in mountainous areas and on shallow limestone plains. Their agricultural potential is limited by their shallow depth and high compaction, which makes them difficult to work on. On the other hand, Regosols are considered very young soils that develop on unconsolidated material, light in color and poor in organic matter. They are common in arid, semiarid (including the dry tropics), and mountainous regions, and they can be found associated to Leptosols and with rock or tepetate outcrops. From the above, it can be deduced that *D. remotiflora* has a great adaptive capacity to poor soils with non-optimal agroclimatic conditions for the rest of the species, which may be an attribute that makes this species an alternative for cultivation in regions where climate change is deteriorating the environmental conditions of agricultural production systems [32].

Regarding soil texture, according to Table 4, *D. remotiflora* prefers coarse-textured soils, typical of Lithosol and Regosol soils, which do not store a large amount of water. Unlike other species of the *Dioscorea* genus, *D. remotiflora* is susceptible to tuber putrescence [9]; therefore, it requires well-drained soils. The presence of *D. remotiflora* detected in medium-textured soils is related to the occurrence of lower annual precipitation levels (735.6 mm on average) than those of sites with coarse-textured soils (1034 mm on average), which ensures that even in soils that store more moisture, it is possible for this species to adapt, as long as the volumes of precipitation are not high; this compensatory effect has been previously reported for diverse crop species [33].

According to Table 4, *D. remotiflora* can tolerate an extreme monthly average minimum temperature of 1.7 °C and, on the other hand, it can survive an extreme monthly average maximum temperature of 41.7 °C, with annual thermal oscillations ranging from 10 to 20 °C and with an optimal annual thermal range of 13 to 16 °C, a fact that shows the wide range of thermal conditions in which this plant can survive, and this includes temperature regimes that are classified as very extreme [34]. Other species of the genus *Dioscorea* have shown tolerance to extreme thermal environments, such as *D. divaricata*, which can tolerate temperatures as low as −18 °C. The occurrence of these extreme minimum temperatures, however, causes delayed full maturity, which is not reached until 3 to 4 years after [35].

On the other hand, extreme temperatures are generally considered a source of dormancy in postharvest tubers and seeds [36]. The dormancy of the tubers of the *Dioscorea* species lasts 120 days, which limits their agricultural production [28]; this indicates an aspect that should be worked on in the immediate future to make *D. remotiflora* a more promising agricultural species. Tuberization is induced by environmental cues such as short days, low temperatures, and higher soil moisture content [36].

Another important aspect in the development of this species is the photoperiod since it intervenes in the formation and growth of leaves and tubers; there are differences among *Dioscorea* species in relation to their response to the photoperiod. In the case of *D. remotiflora*, in the long-day season, which is from May to October, foliar growth is favored, and in short days, the growth and swelling of the tuber are stimulated, promoting the production and storage of starches [37]. In Table 4, it can be observed that the optimal range of the photoperiod for exhibiting adequate leaf growth is from 12.6 to 12.9 h, and for good tuber development, the optimal range is from 11.10 to 11.39 h.

### 3.2. Modeling of Distribution Niches of D. remotiflora

The Kuenm R package implemented with MaxEnt allowed us to obtain an optimal niche model to appropriately depict the *D. remotiflora* distribution in Mexico; based on the requirements of statistical significance, the optimal regularization multiplier, the feature classes, and the omission rate established, the analysis process using Kuenm produced two possible models (Figure 4). One of them was selected, which can be considered a good decision since it fulfills the requirements established [38], and the AUC of the ROC curve accounted for a value greater than 0.93 [39]. The AUC value is an important tool to assess model performance; the higher the AUC value (closer to 1), the better the model performance [40,41]. The Jackknife test identified the most determining variables for the presence of *D. remotiflora*, the precipitation of the warmest quarter, precipitation of the driest month, minimum temperature of the coldest month, November–April solar radiation, annual mean relative humidity, annual moisture availability index, and May–October mean temperature. However, the Jackknife test indicates that the most important variable is solar radiation from November to April (Figure 3). These results agree with what was reported by [42], which mentions that the production of diosgenin, corticosteroids, carbohydrates, and other compounds produced by these species are subject to solar radiation. On the other hand, these results agree with those reported by [43], which pointed out that the environmental variables of major influence for the presence of *Dioscorea humilis* are the precipitation of the wettest quarter, the precipitation of the warmest quarter, and the precipitation of the coldest quarter. On the other hand, [42] reported that for 10 species of the *Dioscorea* genus, the variables that are most important and intervene in the production of secondary metabolites are annual precipitation and average annual radiation.

The capability of *D. remotiflora* to develop in adverse environments observed in the present research could be explained through the ability to adjust its metabolism and modify its morphological characteristics, such as the size and thickness of the leaf, allowing it to adapt to extreme conditions, where other species fail to thrive [44,45].

The current distribution of *D. remotiflora* obtained in the present research (Figure 4) agrees with that reported by [45], which mentions that this species is distributed in Chiapas, Chihuahua, Colima, Durango, Guanajuato, Jalisco, Oaxaca, Guerrero, Michoacán, Guerrero, Nayarit, Puebla, Tamaulipas, Tabasco, and Zacatecas, which represents areas along the foot of the Sierra Madre Occidental, the Sierra Madre del Sur, and its confluence with the Transversal Neovolcanic Axis, where coniferous and oak forests are found; these are home to herbaceous and forest communities with the presence of endemism [46]. However, the distribution, diversity, and structure of populations are strongly influenced by historical, geographical, and climatic events [47], which also trigger speciation processes [48,49]. These arguments explain the number of species of the genus *Dioscorea* and the differences between them in terms of environmental ranges. On the other hand, the areas of high environmental suitability are located on the Mexican Pacific coast; this location suggests that optimal conditions of agroclimatic variables exist in these areas for the development of this plant. In contrast, in the areas of medium and low environmental suitability, it is possible that the species is adapting to agroclimatic ranges that go from very arid to very humid, moving to the central part of Mexico. These areas, therefore, have the potential to establish crops of the species. However, it is important to note that the classification of environmental suitability areas may be modified by the effects of climate change [50]; thus, complementary research regarding potential distribution areas under climate change scenarios would be required. The accessions of *D. remotiflora* found in the northern region of the country and those thriving in the southern part of Mexico may be exhibiting specific adaptations to confront the unique climatic conditions of their respective regions. This observation suggests the existence of local adaptations in response to regional environmental factors, such as variations in temperature, water availability, and other climate-related factors. In the context of *D. remotiflora’s* geographical distribution, it is crucial to consider how populations in different geographic locations may have evolved to cope with specific environmental challenges. Populations in the north may have developed traits that enable them to cope with arid climates and extreme temperatures, while those in the south may have evolved in response to tropical climatic conditions and increased water availability [51]. The adaptation of plant species to specific climatic conditions is a critical phenomenon for their survival and success in various environments. This adaptive variability in *D. remotiflora* could have significant implications for the production of economically important secondary metabolites such as diosgenin, as well as for its cultivation and conservation [42]. Understanding how these regional adaptations influence its ability to thrive in different environmental conditions could guide more effective management and conservation strategies. On the other hand, it is crucial to consider the inherent biases in the algorithms of both ecological niche models and climate change models, as these biases can impact the accuracy and reliability of the results obtained. Understanding and appropriately mitigating these biases are essential elements for enhancing the utility and reliability of these models in decision making related to the conservation and management of plant species and ecosystems [52].

### 3.3. Dioscorea remotiflora Cultivation Prospects

Currently, *D. remotiflora* is a plant that is exploited for human consumption, mainly through the collection of tubers in its natural habitat (Figure 5). However, it presents favorable characteristics for integration into regional crop patterns, given its comparative advantages over other plant species, such as its adaptation and production in poor, shallow, and infertile soils, as well as its tolerance to drought, since it does not have a high water requirement for its development. Some characteristics inherent to the species should be taken into account before considering it as a cultivation option, such as the fact that sexual propagation is not currently considered a viable alternative due to the fact that the seeds it produces are attacked by pests, damaging 60 to 80% of them; thus, the health of the seeds should be ensured through pest control or opting for asexual propagation through the sprouting of tubers. However, the tubers have a prolonged dormancy of up to 120 days, which is a limitation to their propagation. This leads to the need to explore some of the available techniques to reduce the duration of dormancy and accelerate the germination process. Nevertheless, the conservation of wild species is a global challenge that demands coordinated efforts at the local, national, and international levels. Each species and situation may necessitate specific approaches, such as monitoring and tracking, education and awareness, legislation and regulation, international collaboration, in situ and ex situ conservation programs, research and technology, community participation, and sustainable development, which can serve as a starting point for designing effective conservation strategies [53].

## 4. Materials and Methods

### 4.1. Occurrence Data

During the research process, a total of 1030 geo-referenced accessions were identified and collected from various sources, including herbaria, floristic inventories, scientific articles, and databases. Subsequently, a thorough review of these records was conducted to eliminate duplicates, records with incorrect geographic coordinates, and those located outside the study area [54]. These rigorous strategies were meticulously employed to ensure that only accessions belonging to the actual natural distribution areas of interest were considered [55]. As a result of this selection process, the sample was reduced to a set of 480 records, detailed in Table 6.

### 4.2. Climatic Data

Based on previous studies for species of the same genus [42], monthly, quarterly, seasonal, and annual rasters of precipitation, temperature, solar radiation, and relative humidity were used to determine potential distribution areas of *D. remotiflora.* These climatic data were obtained from the Agroclimatic Information System for México and Central America (SIAMEXCA) [21]. The raster images have a resolution of 30” of arc and correspond to the period 1961–2010. From the SIAMEXCA rasters, other additional variables were generated, adding a total of 42 environmental variables (Table 7).

### 4.3. Environmental Characterization of the Occurrence Sites

Based on the geographic coordinates of the *D. remotiflora* occurrence sites, an environmental characterization of these sites was carried out using the 42 variables mentioned in Table 2. For this, data extraction procedures were conducted with the raster images, using the ArcMap software version 10.8 [56]. With data extracted from the 42 variables, an environmental data matrix (EDM) was built in Microsoft Excel.

### 4.4. Selection of Environmental Variables

Prior to the execution of the statistical analysis, the Shapiro–Wilk test was applied to verify the data’s normality, not finding normality (*p* < 0.05) for any of the data series within the 42 environmental variables included in the EDM.

Diverse studies have shown that multicollinearity is a problem that can cause high correlations among independent variables, a fact that can lead to unreliable and unstable estimations of the regression coefficients [57]. To determine the presence of multicollinearity, a Spearman correlation coefficient r > 0.9 was established as a threshold value [38]. In this way, the correlated variables with an r < 0.9 coefficient were selected, and among the variables with collinearity, the one considered the most relevant for the presence of the species was selected. Data from the EDM were used to perform the correlation analyses; these statistical analyses were carried out with programs developed in the R software, version 4.05 [58], and with normalized data. The results of these analyses reported 20 useful variables; later, preliminary analyses were carried out in MaxEnt individually and in conjunction with the Kuenm R package in order to use the Jackknife test to select the most relevant variables in the distribution of *D. remotiflora* and thus carry out the final modeling of the ecological niche.

### 4.5. Characterization of the Adaptive Capacity of D. remotiflora

Based on the geographic location of the *D. remotiflora* occurrence sites, the agroclimatic regions, soil units, and vegetation types where the species is currently distributed were characterized. For this purpose, a map of agroclimatic regions of Mexico and Central America was used [21], as well as a soil unit map and a vegetation type map for Mexico [59]. This made the elaboration of a list of the edaphoclimatic conditions to which *D. remotiflora* currently adapts to and its preference for certain habitats possible [60].

In addition, the ecological descriptors for *D. remotiflora* were determined using the EDM information but only taking into account the 20 environmental variables that were selected after the correlation–collinearity analysis. The ecological descriptors were established in terms of environmental ranges for *D. remotiflora* adaptation and optimal environmental ranges for *D. remotiflora* presence, which corresponded to the highest frequency of occurrence sites.

### 4.6. Ecological Niche Modeling

Ecological niche modeling was performed using the MaxEnt model, which uses the principle of maximum entropy with species presence and environmental data to create a correlative model that relates the ecological requirements of a species with the regional environmental availabilities to predict the relative habitat suitability. Also, it allows us to derive specific descriptors to enrich the ecological characterization of the territories [61,62,63]. We used the Kuenm R package [18,58,64] to automate and optimize the ecological niche modeling (ENM) process. Kuenm’s Kuenm_ceval function creates preliminary models [63] with occurrence data and environmental predictors and also evaluates the efficiency of these models through their statistical significance with the cal_eval function; in addition, the relative quality of the model is evaluated by using the Akaike Information Criterion (AIC), corrected for small sample sizes (AICc) [18]. Kuenm also enables us to assess diverse regularization multiplier (RM) factors, combinations of feature classes (FCs), and different groups of environmental predictors, as well as to establish the allowable omission rate (OR). For each parameter setting, two models are created: one based on the complete set of occurrences and the other based only on the training data. Thus, the final niche model was selected according with the criteria sequence shown in Table 8.

For this research, models were tested using a sequential order of the FCs (L, LQ, H, LQH, LQHP, LQHPT) and RM values of 0.1 to 5 with 0.1 increases, a maximum omission rate of 5%, and 50 k-fold replicates of each configuration; 500 iterations were used [64]. The *D. remotiflora* occurrence sites and the ASCII files for the 20 variables selected after correlation–collinearity analysis were used as inputs in the ENM process with the Kuenm R package.

## 5. Conclusions

*D. remotiflora* is an endemic species from Mexico whose tubers are collected in their natural habitat for human consumption. *D. remotiflora* is mostly adapted to semiwarm to warm environments and to semiarid to subhumid climates with a long dry season; thus, it could be considered as a good crop option for environments with drought and heat episodes. Moreover, considering that *D. remotiflora* has a greater presence in regions with very thin, stony, poorly developed, and low-fertility soils. All these characteristics make *D. remotiflora* a species with comparative advantages to develop in edaphoclimatic environments that are adverse for most regional crop species.

*D. remotiflora* is currently distributed in the western portion of Mexico, along regions bordering the Mexican Pacific, which hints at its center of geographic origin. Niche modeling identified precipitation of the warmest quarter, precipitation of the driest month, mean minimum temperature of the coldest month, November–April mean solar radiation, annual mean relative humidity, annual moisture availability index, and May–October mean temperature as the variables with the greatest contribution to explaining the presence of *D. remotiflora*. Furthermore, the Kuenm R package enabled the selection of a niche model that optimized the depiction of the potential distribution areas for this species. Thus, potential areas with high environmental suitability for *D. remotiflora* were located in the Mexican States where this species is already present, and potential areas with low-to-medium environmental suitability were identified in regions with the current presence of *D. remotiflora* as well as regions where it has not yet been reported, such as in the Yucatán Peninsula, northeast region, and the Gulf of Mexico.

The cultivation perspectives of *D. remotiflora* are favorable considering its capacity to adapt to harsh environments and that its nutritional and medicinal properties are valuable; however, the prolonged dormancy of its tubers is one of the intrinsic aspects of this plant that should be overcome in the near future to ease its incorporation into regional crop patterns.

Currently, plant genetic resources worldwide are facing the pressure of overexploitation and environmental change, resulting in habitat fragmentation and biodiversity threats. To ensure the permanence of these natural resources, it is essential to maintain or increase the resilience of these systems against these pressures. Therefore, an efficient conservation management model must be adopted to address these changes and adequately preserve *D. remotiflora* populations.

## Figures and Tables

**Figure 1 plants-12-03654-f001:**
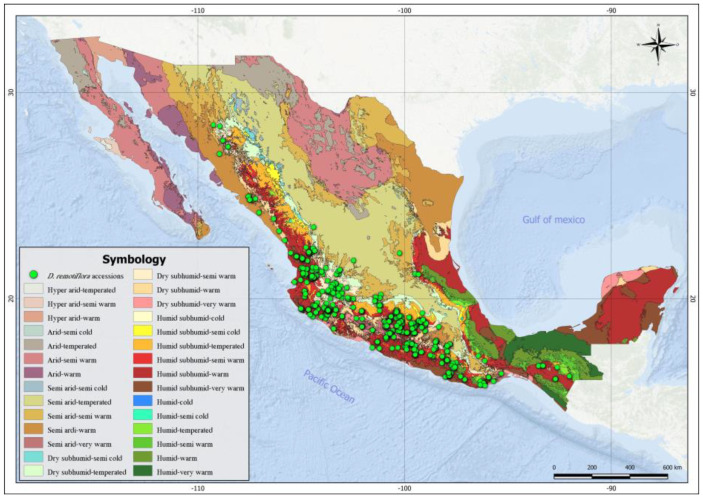
The current distribution of *D. remotiflora* across the agroclimatic regions of Mexico.

**Figure 2 plants-12-03654-f002:**
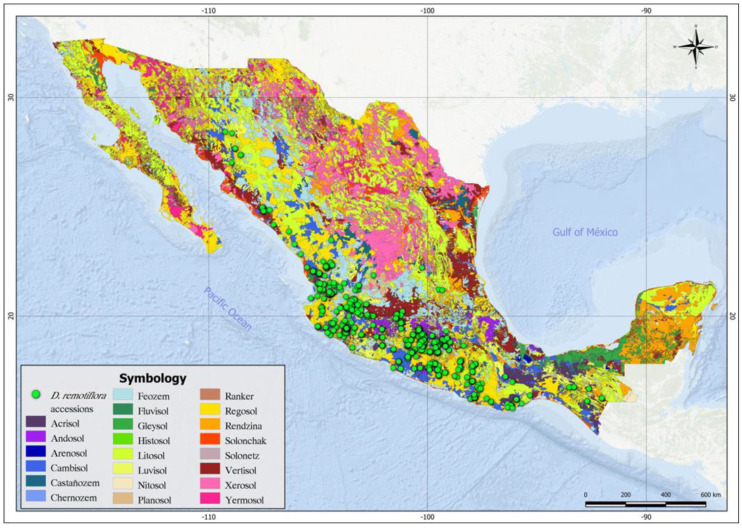
Current distribution of *D. remotiflora* across soil units of Mexico.

**Figure 3 plants-12-03654-f003:**
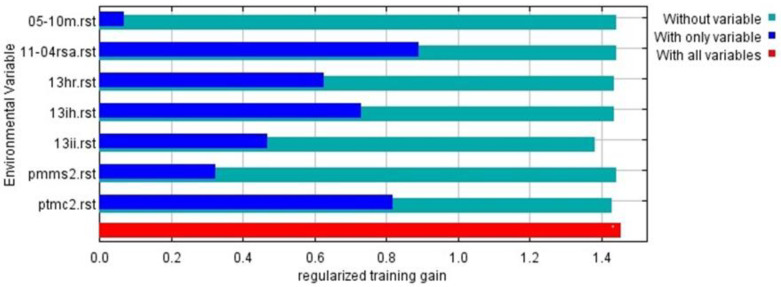
Results of Jackknife test of the relative importance of predictor environmental variables in MaxEnt model for *D. remotiflora* in Mexico.

**Figure 4 plants-12-03654-f004:**
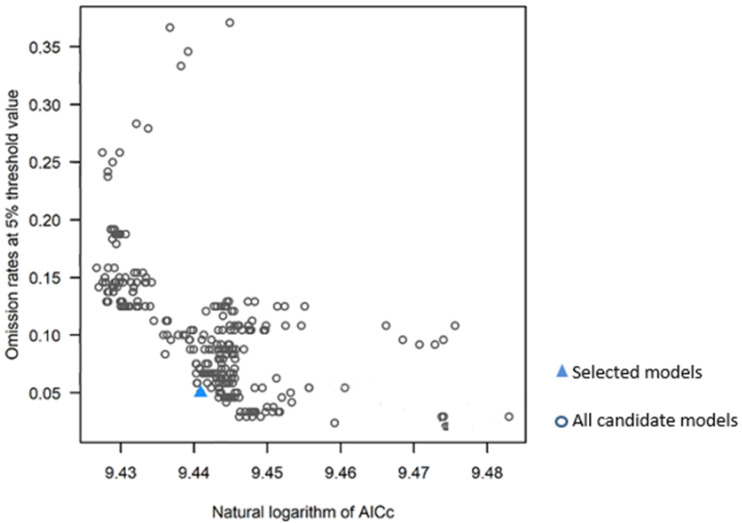
Models obtained and evaluated by Kuenm R package for *D. remotiflora*.

**Figure 5 plants-12-03654-f005:**
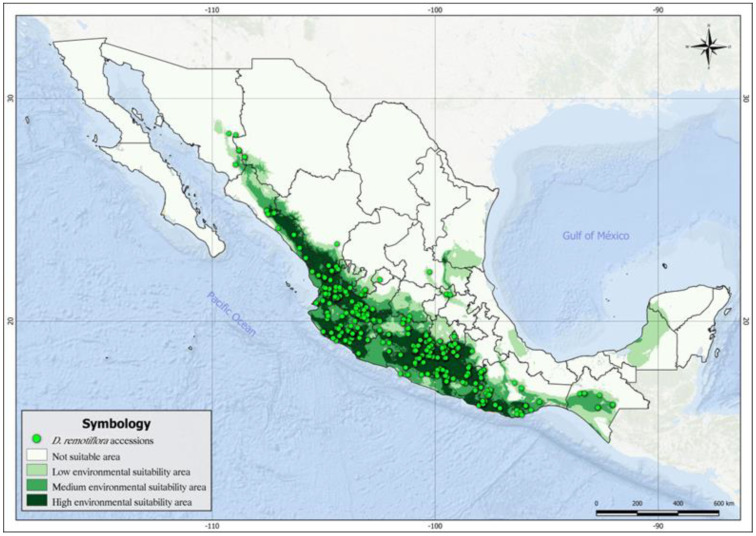
Areas with environmental suitability for *D. remotiflora* in Mexico.

**Table 1 plants-12-03654-t001:** Annual mean temperature and annual moisture availability index intervals for 17 agroclimatic regions with the presence of *D. remotiflora* in Mexico.

Agroclimatic Region	Annual Moisture Availability Index	Annual Mean Temperature (°C)	Total Accessions
Semiarid very warm	0.2–0.5	>26	17
Semiarid warm	0.2–0.5	22–26	22
Semiarid semiwarm	0.2–0.5	18 a 22	14
Semiarid temperate	0.2–0.5	12 a 18	3
Dry–subhumid very warm	0.5–0.65	> 26	20
Dry–subhumid warm	0.5–0.65	22–26	66
Dry–subhumid semiwarm	0.5–0.65	18–22	88
Dry–subhumid temperate	0.5–0.65	12–18	7
Humid–subhumid very warm	0.5–0.65	>26	19
Humid–subhumid warm	0.65–1.0	22–26	67
Humid–subhumid semiwarm	0.65–1.0	18–22	98
Humid–subhumid temperate	0.65–1.0	12–18	11
Humid very warm	>1.0	>26	2
Humid warm	>1.0	22–26	26
Humid semiwarm	>1.0	18–22	12
Humid temperate	>1.0	12–18	6
Humid semicold	>1.0	5–12	3

**Table 2 plants-12-03654-t002:** Soil units and soil texture classes with the presence of *D. remotiflora*.

FAO Soil Unit	Soil Texture	Total Accessions
Lithosol	Coarse	108
Regosol calcaric	Coarse	57
Regosol eutric	Coarse	209
Faozem haplic	Coarse	34
Vertisol cromic	Fine	39
Solonchak ortic	Fine	22
Fluvisol eutric	Medium	10
Fluvisol calcaric	Coarse	1

**Table 3 plants-12-03654-t003:** Ecological descriptors for *D. remotiflora*.

Environmental Variables	Min	Max	Optimum
1. Precipitation of the warmest quarter (mm)	240	1204	400–884
2. Precipitation of the driest month (mm)	1	73	1–7
3. Annual mean precipitation (mm)	444	2886	700–1299
4. May–October mean precipitation (mm)	344	1943	700–1199
5. November–April mean precipitation	23	863	30–100
6. Annual moisture availability index	0.27	2.32	0.40–0.99
7. November–April availability index	0.026	1.83	0.030–1300
8. May–October availability index	0.005	1.47	0.009–1.4
9. Maximum maximorum temperature (°C)	24.61	41.17	29–37
10. Minimum minimorum temperature (°C)	1.7	18.2	5–15
11. Annual mean temperature (°C)	14.66	28.51	19–27
12. May–October mean temperature	9.13	29.88	19–26
13. November–April mean temperature	7.95	27.83	19–26
14. Annual thermal oscillation (°C)	10.42	19.54	13.16
15. Annual temperature range (°C)	1.54	14.52	3–7
16. Soil texture	Sandy	Fine	Medium
17. May–October mean photoperiod (h)	12.5	12.9	12.6–12.9
18. November–April mean photoperiod (h)	10.97	11.47	11.10–11.39
19. Growing season			120–190
20. Altitude (mm)	6	4295	200–1800

Min = minimum value; Max = maximum value; Optimum = optimal range

**Table 4 plants-12-03654-t004:** Contribution of seven environmental variables to determining the presence and distribution of *D. remotiflora*.

Environmental Variables	Contribution (%)	Permutation Importance (%)
Precipitation of the warmest quarter (mm)	42.4	49
Precipitation of the driest month (mm)	17.5	2.8
Minimum temperature of the coldest month (°C)	15	31
November–April mean solar radiation (w/m^2^)	10	0.8
Annual mean relative humidity (%)	8.5	3.6
Annual moisture availability index	5.7	7.4
May–October mean temperature (°C)	0.9	5.3

**Table 5 plants-12-03654-t005:** Characteristics of the model selected to depict the potential distribution of *D. Remotiflora* in Mexico.

Parameter	Value
AUC of the ROC curve	0.935
Mean AUC ratio	1.679
Omission rate (%)	0.5
AICc	12,592.861
Delta AICc	0
W AICc	0.9999
Optimum regularization multiplier	3.0

**Table 6 plants-12-03654-t006:** Occurrence data sources for *D. remotiflora*.

Institution/Source	Institution/Department	Accessions
Universidad Nacional Autónoma de México (UNAM).	Instituto de Biología	169
Instituto de Ecología (INECOL).	Xalapa Veracruz	30
Universidad Autónoma de Querétaro (UAQ)	Facultad de Ciencias Naturales	3
Instituto Nacional de Estadística y Geografía (INEGI).	Departamento de Botánica	2
Universidad Autónoma de Aguascalientes (UAA).	Centro de Ciencias Básicas	2
Universidad Autónoma de Veracruz (UPAV) (CIB).	Instituto de Investigaciones Biológicas	1
Universidad Autónoma de San Luis Potosí (UASLP).	Instituto de Investigación de Zonas Desérticas	3
Colegio de la Frontera Sur (ECO SUR).	Herbario San Cristóbal	3
Universidad de Guadalajara (CUCBA, CUC SUR).	Herbario IBUG, Herbario ZEA	6
Universidad Michoacana de San Nicolás de Hidalgo, Morelia, Michoacán.	Herbario Facultad de Biología Universidad Michoacana	6
Artículos científicos/Inventarios florísticos de los estados de Oaxaca, Chiapas, Veracruz, Tabasco, Guerrero, Puebla, Jstor Plant Science.		20
Universidad Autónoma de Nuevo León (UNL).	Facultad de Ciencias Biológicas	1
La Comisión Nacional para el Conocimiento y Uso de la Biodiversidad (CONABIO).	Herbario digital de CONABIO	3
Trópicos.org.		3
Red de Herbarios del Noroeste de México.		13
GBIF		215
Total		480

**Table 7 plants-12-03654-t007:** Environmental variables considered in this research.

Variable	Description	Temporal Scale
BIO01	Annual mean temperature	Annual
BIO02	Mean diurnal range	Variation
BIO03	Isothermality	Variation
BIO04	Temperature seasonality	Variation
BIO05	Maximum temperature of the warmest month	Month
BIO06	Minimum temperature of the coldest month	Month
BIO07	Temperature annual range	Annual
BIO08	Mean temperature of the wettest quarter	Quarter
BIO09	Mean temperature of the driest quarter	Quarter
BIO10	Mean temperature of the warmest quarter	Quarter
BIO11	Mean temperature of the coldest quarter	Quarter
BIO12	Annual precipitation	Annual
BIO13	Precipitation of the wettest month	Month
BIO14	Precipitation of the driest month	Month
BIO15	Precipitation seasonality	Variation
BIO16	Precipitation of the wettest quarter	Quarter
BIO17	Precipitation of the driest quarter	Quarter
BIO18	Precipitation of the warmest quarter	Quarter
BIO19	Precipitation of the coldest quarter	Quarter
N-AMT	November–April mean temperature	Seasonal
M-OMT	May–October mean temperature	Seasonal
M-OXT	Maximum temperature May–October	Seasonal
N-AXT	November–April maximum temperature	Seasonal
AXT	Annual maximum temperature	Annual
M-OIT	May–October minimum temperature	Seasonal
N-AIT	November–April minimum temperature	Seasonal
AIT	Annual minimum temperature	Annual
ATO	Annual thermal oscillation	Annual
M-OP	May–October precipitation	Seasonal
N-AP	November–April precipitation	Seasonal
M-OPH	May–October photoperiod	Seasonal
N-APH	November–April photoperiod	Seasonal
AMI	Annual moisture index	Annual
M-OMI	May–October mean moisture index	Seasonal
N-AMI	November–April mean moisture index	Seasonal
ASR	Annual mean solar radiation	Annual
M-OSR	May–October mean solar radiation	Seasonal
N-ASR	November–April solar radiation	Seasonal
ARH	Annual relative humidity	Annual
M-ORH	May–October relative humidity	Seasonal
N-ARH	November–April relative humidity	Seasonal
GSL	Growing season length	Seasonal

BIO = bioclimatic variable.

**Table 8 plants-12-03654-t008:** Selection criteria to select the optimal ecological niche model.

Criteria
All candidate models
Statistically significant models
Models meeting omission rate criteria
Models meeting AICc criteria
Models meeting high AUC value
Statistically significant models meeting omission rate criteria
Statistically significant models meeting AICc criteria
Statistically significant models meeting high AUC value
Statistically significant models meeting omission rate criteria, AICc criteria, and AUC criteria

## Data Availability

Not applicable.

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
