# Peer review of "Ecogeography of *Dioscorea remotiflora* Kunth: An Endemic Species from Mexico"

_plants, 2023, doi:10.3390/plants12203654_

Round 1
Reviewer 1 Report
Dear Authors,
The paper titled "Ecogeography of Dioscorea remotiflora Kunth reveals its potential in regions already affected by climate change" aims to investigate the ecogeography and potential distribution of Dioscorea remotiflora, a native species of Mexico. The authors discuss the ecological and nutritional characteristics of this species and emphasize the need for research to facilitate its conservation and utilization. They use occurrence data to conduct an eco-geographical analysis and develop models of its potential geographic distribution.
Overall, this paper addresses an important topic related to the ecological understanding and potential utilization of an underutilized plant species, especially in the context of climate change and food production challenges. However, there are several aspects that need attention and improvement:
Abstract:
- The abstract should be revised to ensure it provides a concise summary of the study's key findings, methodology, and implications.
- It should explicitly mention the major environmental variables that influenced the distribution of D. remotiflora.
- Consider including a statement about the broader implications of the research for conservation and cultivation strategies.
Introduction:
- Clarify the significance of studying D. remotiflora's distribution and adaptive capacity, including any ecological or economic importance.
- Provide a more focused and concise introduction to set the context for the research.
- State clear research objectives and hypotheses in the introduction.
Materials and Methods:
- Ensure that the data collection and processing procedures, such as the selection criteria for accessions and the elimination of repeated records, are described in detail.
- Explain how the 42 environmental variables were chosen, their relevance, and any preprocessing steps in the materials and methods section.
- Provide a clear rationale for the selection of specific environmental variables and how they relate to the species' ecology.
- Describe the specifics of the statistical analysis, including the parameters used in the MaxEnt model, within the materials and methods section.
Results:
- In the results section, it's important to highlight the most important findings, such as the key environmental variables affecting D. remotiflora's distribution.
- Provide detailed information on the model's performance, including evaluation metrics and validation procedures in the results section.
Discussion:
- Interpret the results in the context of the study's objectives and broader ecological knowledge in the discussion.
- Discuss the ecological implications of the species' adaptive capacity in different environments within the discussion.
- Consider addressing any limitations or potential sources of bias in the study, such as data availability or selection criteria, within the discussion.
- Provide insights into how the research findings can contribute to the conservation and cultivation of D. remotiflora in the discussion.
Conclusion:
- In the conclusion, summarize the main findings concisely.
- Offer clear recommendations or implications for conservation and cultivation strategies in the conclusion.
- Avoid introducing new information or concepts in the conclusion that were not discussed in the main body of the manuscript.
The language and writing style of the manuscript are clear and generally well-structured, making it accessible to readers. However, some minor grammatical and typographical errors need correction for a smoother reading experience.
Author Response
Thanks for your observations.

Reviewer 2 Report
Comments for Author,
I have read your paper carefully. Unfortunately, I can not suggest this paper for the publication because of lack of enough novelt content, poorly quality of introduction and discussion.
- The used references are unproperly. So, Introduction is needed rewrite.
- The tables are so complicated. Therefore, some of them make no sense.
- The figures are unclear. Please use more clear figures. Otherwise, these figures are simple.
- Even though the obtained results are fine, the discussion of paper is superficial.
- Conclusion remark is missed.
Author Response
Thancks for your observations.

Reviewer 3 Report
This is a very interesting and useful work on Dioscorea remotiflora Kunth, a wild plant species that is native to Mexico and serves as an important ethnobotanical and nutritional component of the local flora.
First, I do not like the title a bit (i.e., the 'already affected' wording) because it suggests that some regions of Mexico are affected by climate change and some are not. However, it seems that all regions of Mexico are affected by climate change to a similar extent. Thus, I suggest rewriting the title.
However, the results of these analyses can be implemented in the future, especially in the context of climate change, i.e., in Mexico.
The analyses were performed according to art; the most important fact is that according to the Spearman correlation analysis, 42 environmental variables were reduced to 20 to be used in further analyses. Also the methods used are described very thoroughly.
I really like that the authors managed to determine potential habitats for this plant that could be inhabited in the future. Thorough analyses were carried out in terms of various ecological factors, including even the types of substrates/soils. Also Subsection 3.3 (D. remotiflora cultivation prospects) is very useful (however, the title of the subsection should not begin with the abbreviation of the generic name; please develop to the full generic name).
I actually only have minor comments related to the occurrence of minor typos (or for grammatical inconsistencies) in the text or some others:
Table 6, should be: Precipitation instead of Pprecipitation.
One free line should be inserted before the titles of chapters and sub-chapters (if it is missing).
Line 44: I understand that it is a species endemic to Mexico, but it must also be clearly written and should not be guessed from the context of this sentence.
Line 294: The sentence cannot start this way: '[28] mentions...'. The name of the researcher(s) is needed there.
The spelling on geographic names and other nouns is inconsistent; sometimes they are provided in Spanish (e.g., México instead of Mexico, throughout the manuscript; or hábitats vs. habitats, in line 419, where the square bracket at the end of the sentence is also missing; line 352: áreas is also in Spanish), sometimes in English (Yucatan instead of Yucatán, e.g., line 33). I think the authors strongly need to decide on some unification of the spelling through the MS.
The compatibility of the singular and plural in the sentences should be checked because currently some parts of the sentences are incompatible with each other (e.g. in line 45 ‘roots, tubers, and rhizomes’ and yet in line 47 the rest of the sentence is singular).
There are different spellings of the same word in the MS: Maxent vs. MaxEnt; or Jackknife test vs. jackknife test; or km2 vs. Km2, and even in one sentence (lines 193-194). Please uniform all such inconsistencies through the MS.
Line 448: ‘whose’, i.e., ‘belonging to or associated with which person’ but the plant mentioned before is not the case.
The tables are quite extensive and take up, for example, two or three pages. If the huge spaces between the lines were removed, they would certainly take up much less space and be more readable for the reader.
Author Response
Thancks for your observations

Round 2
Reviewer 2 Report
Author improved paper carefully. Now, the paper could be accepted for the publication.
Best Regards